# Pervasive mRNA uridylation in fission yeast is catalysed by both Cid1 and Cid16 terminal uridyltransferases

Lidia Lipińska-Zubrycka[1], Maciej Grochowski[1], Jürg Bähler[2], Michał Małecki[1]*

**1** Institute of Genetics and Biotechnology, Faculty of Biology, University of Warsaw, Warsaw, Poland,
**2** Institute of Healthy Ageing and Research Department of Genetics, Evolution & Environment, University College London, London, United Kingdom

* m.malecki@igib.uw.edu.pl

**Data Availability Statement:** Raw ".fastq" files from RNA sequencing were deposited in in Sequence Read Archive (SRA, NCBI) under accession number: PRJNA862652. The raw data

## Abstract

Messenger RNA uridylation is pervasive and conserved among eukaryotes, but the consequences of this modification for mRNA fate are still under debate. Utilising a simple model organism to study uridylation may facilitate efforts to understand the cellular function of this process. Here we demonstrate that uridylation can be detected using simple bioinformatics approach. We utilise it to unravel widespread transcript uridylation in fission yeast and demonstrate the contribution of both Cid1 and Cid16, the only two annotated terminal uridyltransferases (TUT-ases) in this yeast. To detect uridylation in transcriptome data, we used a RNA-sequencing (RNA-seq) library preparation protocol involving initial linker ligation to fragmented RNA—an approach borrowed from small RNA sequencing that was commonly used in older RNA-seq protocols. We next explored the data to detect uridylation marks. Our analysis show that uridylation in yeast is pervasive, similarly to the one in multicellular organisms. Importantly, our results confirm the role of the cytoplasmic uridyltransferase Cid1 as the primary uridylation catalyst. However, we also observed an auxiliary role of the second uridyltransferase, Cid16. Thus both fission yeast uridyltransferases are involved in mRNA uridylation. Intriguingly, we found no physiological phenotype of the single and double deletion mutants of *cid1* and *cid16* and only minimal impact of uridylation on steady-state mRNA levels. Our work establishes fission yeast as a potent model to study uridylation in a simple eukaryote, and we demonstrate that it is possible to detect uridylation marks in RNA-seq data without the need for specific methodologies.

## Introduction

Messenger RNA (mRNA) degradation has a critical role in regulating transcript levels and is a major point of gene expression regulation. Bulk cytoplasmic mRNA decay was most studied in budding yeast, where it is initiated by poly(A) tail shortening followed by removal of the 5' cap (decapping) [1]. Removal of this protective cap structure renders mRNA accessible for the cytoplasmic 5'-3' exonuclease Xrn1. In yeast, 5'-3' decay is considered the main degradation

file containing reads with 3'-A/U extensions was deposited in OSF database - https://osf.io/em7vp/.

**Funding:** This work was funded by EU Operational Programme Innovative Economy via the Foundation for Polish Science grant FIRST TEAM awarded to MM (POIR.04.04.00-00-4316/17). This work was supported by Wellcome Senior Investigator Award to JB (095598/Z/11/Z). This work was supported by National Science Center MIniatura grant (2019/03/X/NZ2/00787) awarded to LL-Z. The funders had no role in study design, data collection and analysis, decision to publish, or preparation of the manuscript.

**Competing interests:** the authors have declared that no competing interests exist.

pathway, however, mRNA can be also be degraded from the 3' end by the exosome complex supported by the SKI complex [2]. While the degradation machinery is evolutionary conserved, progress in analysing other eukaryotic models demonstrated some peculiarities of the budding yeast system [3]. Most importantly, while pervasive cytoplasmic uridylation of mRNAs was detected in most studied eukaryotes it is absent in budding yeast [4, 5].

Uridylation is catalysed by cytoplasmic terminal uridyltransferases (TUTases) [6, 7]. It is believed that transcript uridylation induces its degradation either by attracting the LSM complex which accelerates decapping and subsequent 5'-3' decay or by triggering 3'-5' decay by Dis3l2 exonuclease that preferentially targets uridylated RNAs [4]. Uridylation was found to play important role in specific processes like antiviral response [8], transposon control [9], germline development [10, 11], or RNA degradation triggered by apoptosis [12]. However, the significance of pervasive mRNA uridylation in the context of bulk mRNA turnover is unclear, and elimination of uridylation generates no detectable consequences for the physiology of somatic cells and only limited molecular manifestations [7, 11, 13].

Messenger RNA uridylation was first detected in fission yeast *(Schizosaccharomyces pombe)* [5], another well studied single-cell eukaryotic model evolutionary distant from budding yeast. The fission yeast genome encodes two TUT-ase genes, *cid1* and *cid16* [14]. Cid1 was shown to be the main enzyme responsible for mRNA uridylation, while the contribution of Cid16 to this process is not known [5]. The *S. pombe* U-tail-specific exonuclease Dis3l2 was shown to genetically interact with mRNA decay factors suggesting its contribution to the bulk degradation process [7]. Uridylation in fission yeast was so far only detected on a few tested transcripts [5, 7, 15].

Currently, the only described method for genome-wide uridylation analysis is TAIL-seq. TAIL-seq involves specific library preparation that due to adapter ligation preserves 3'-end information; moreover, the TAIL-seq method involves a bioinformatics pipeline that improves the base-calling of the sequences including long adenine homopolymer [16]. TAIL-seq brought numerous new insights to our understanding of the eukaryotic mRNA degradation process, but it is a technically and computationally challenging method.

Some of the standard RNA sequencing library preparation protocols involving 3'-adapter ligation to RNA after initial fragmentation also preserve some 3'-end information. This kind of approach is commonly used in small RNA library preparation protocols and can be adapted to long RNAs sequencing, in such case adapter is ligated after RNA fragmentation. In this case 3'-end sequence can be used to map polyadenylation sites [17]. Here we used a fission yeast model to demonstrate that also uridylation can be detected in such data with simple analysis pipeline. Our results show pervasive mRNA uridylation in fission yeast that shares characteristics with uridylation detected in higher eukaryotes. Moreover, we observed that both Cid1 and Cid16 uridyltransferases contribute to mRNA uridylation. Furthermore, we show that uridyltransferases deletion does not induce any detectable cell growth phenotypes and has only a limited impact on mRNA steady-state levels.

## Results

### Detecting 3'-end uridylation in RNA-seq reads

One way of RNA sequencing library preparation is through ligation of adapters to fragmented mRNA before library amplification. Sequencing data obtained from such libraries contain some reads with full or fragmented poly(A) tails, and such reads can be used to map the poly (A) sites of transcripts [17]. We reasoned that terminal uridylation could also be detected in this type of data (Fig 1A). To check our idea, we prepared RNA sequencing libraries by ligating adapters to rRNA-depleted and fragmented total RNA using a previously described protocol [18]. Libraries were then subjected to Illumina sequencing, and the reads were aligned to the

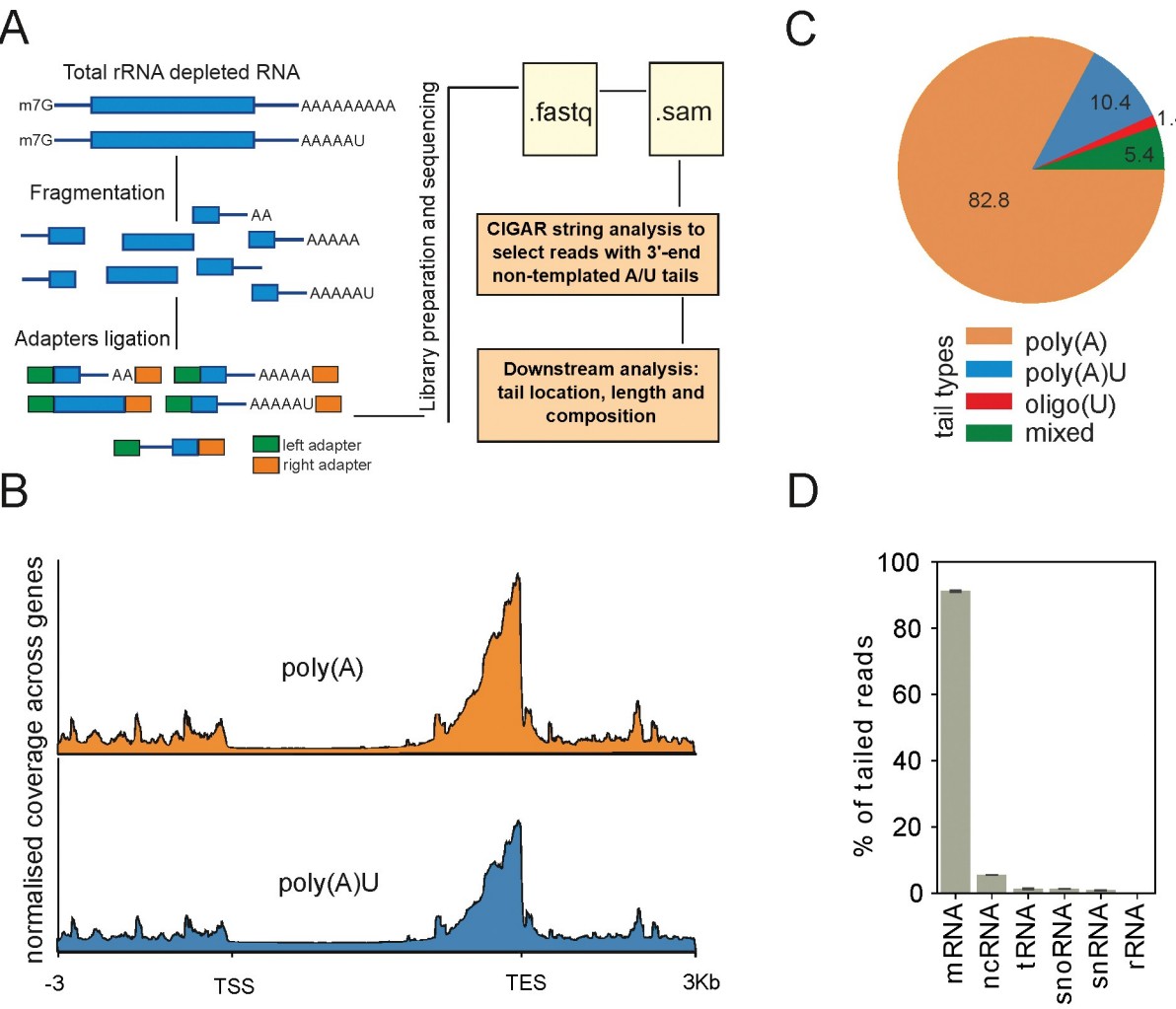

**Fig 1. Identification of uridylation in RNA-seq data. (A)** Scheme of workflow used to identify uridylation. Libraries prepared by ligating adapters to fragmented RNA contain 3'-end uridylated reads. We searched for such reads using information encoded in the ".sam" file CIGAR string. **(B)** The majority of detected reads with un-templated A/U extensions map closely to annotated transcription end sites (TES). **(C)** Using our approach to search for reads with A/U un-templated extensions we detected reads with pure poly(A) or poly(A) extensions additionally uridylated at the 3'-end—poly(A)U. Additionally, some reads had pure U tails—oligo(U) or did not fall into any of those categories—mixed. The percent fractions of detected reads are indicated as numbers on the pie chart. **(D)** The majority of detected reads with un-templated A/U extensions longer than 3nt mapped to mRNAs (average of two repeats of wild type sample is shown).

genome using an aligner with standard soft-clipping options (see Methods). Next, we used information encoded in the ".sam" file CIGAR string to identify reads with 3'- end non-templated nucleotides and extract tail sequence (Fig 1A). We next searched extracted tails for ones composed of adenines (As) and uridines (Us). We have assigned extracted tails to the following categories: poly(A), poly(AU) or oligo(U) or somewhat mixed A and U order (mixed). To minimise potential artefacts due to sequencing errors, we focused on the reads with extensions being at least 4 nucleotides long. The fraction of identified reads with 3'-end adenine and/or uridine un-templated extensions accounted for approximately 1.5% of total aligned reads (around 20 thousand reads for wild type samples).

As expected, the identified reads mapped to the 3'-end of RNAs (Fig 1B). In agreement with our hypothesis in the wild-type strain, we detected reads with poly(A) and poly(AU) extensions. Those were two dominant categories of detected tailed reads, accounting for 93%

of all A/U tailed reads (Fig 1C). Both poly(A) and poly(AU) reads mapped at the 3'-end of RNA confirming that we observe predominantly uridylation of pre-existing poly(A) tails (Fig 1B). Oligo(U) reads were detected at a much lower frequency (Fig 1C). As expected, the detected tailed reads aligned predominantly to mRNAs, and we further focused our analysis on this category (Fig 1D).

## Both Cid1 and Cid16 participate in the uridylation of *S. pombe* mRNA

To confirm that observed uridylation originates from the enzymatic activity of terminal uridyltransferases, we created deletion strains lacking either or both *cid1* and *cid16*, encoding the known fission yeast TUT-ases. Sequencing libraries were prepared in the same way as for the wild-type cells, followed by detecting reads with the 3'-end A/U non-templated extension in the sequencing data. In agreement with uridyltransferases function, the detected mRNA uridylation decreased in TUT-ase deletion strains (Fig 2A). This result confirms that our data reflect the underlying biology and validates our method of detecting uridylation.

The *cid1* deletion resulted in the loss of 51.9% of uridylation compared to the wild-type cells (11% uridylated reads in wild-type in the comparison with 5.3% in Δ*cid1*). Deletion of the *cid16* gene alone had no significant impact on the uridylation frequency. Notably, deletion of both *cid1* and *cid16* resulted in further loss of uridylation (81.1% loss compared to the wild-type cells) (Fig 2A). Therefore, our result confirms the role of Cid1 as the main cytoplasmic uridyltransferase acting on mRNA poly(A) tails, but also show that Cid16 can participate in cytoplasmic mRNA uridylation.

According to the literature, uridylation occurs at the 3'-end of shortened poly(A) tails and consists predominantly of 1 or 2 uridine residues [4, 5, 7, 16]. In agreement with this, most of the detected reads with poly(A) and poly(AU) tails mapped closely to annotated transcription end sites (TES), while this was not as clear for oligo(U) tails (Fig 2B). Among uridylated reads detected in our data, the majority contained one or two uridines added to poly(A) tails (Fig 2C). We noticed that while poly(A) tailed reads mapped to TES in all analysed mutants, the poly(AU) reads detected in the double mutant (Δ*cid1*Δ*cid16*) largely lost this correlation. This result suggests a possible artefactual origin of the detected residual reads indicating a lack of uridylation in the absence of both annotated uridyltransferases (Fig 2D).

## Uridylation is pervasive in fission yeast

We next looked at individual transcripts for which we detected reads with non-templated nucleotide extensions. We took into account wild-type (WT) and Δ*cid16* strains where uridylation was detected at high frequency. To increase gene number for analysis we added counts that were recorded for two biological repeats of each sample, next we considered transcripts for which sum of at least 5 tailed poly(A) or poly(A)U reads were detected. This threshold resulted in 940 transcripts in the wild-type samples and 1300 in the Cid16 deletion strain samples (Fig 3A). For roughly 70% of the transcripts we detected both poly(A) and poly(AU) read categories. There were almost no transcripts with reads bearing only poly(A)U extensions (1 transcript in WT and Cid16 sample). Additionally, the average read number for transcripts with only poly(A) tailed reads was significantly lower than average read number for the sample (source data in S1 Table). All this suggests that uridylation is a common feature of all transcripts, and lack of detection is solely the result of a low read number.

Accordingly, we observed that the number of reads with A/U extension positively correlated with the read number detected for a given transcript (Fig 3B and 3E). Using the *cid16* deletion sample as an example (due to higher sequencing depth it had more genes passing our read number threshold), we observed that transcripts for which we detected at least 5 reads with polyA/AU

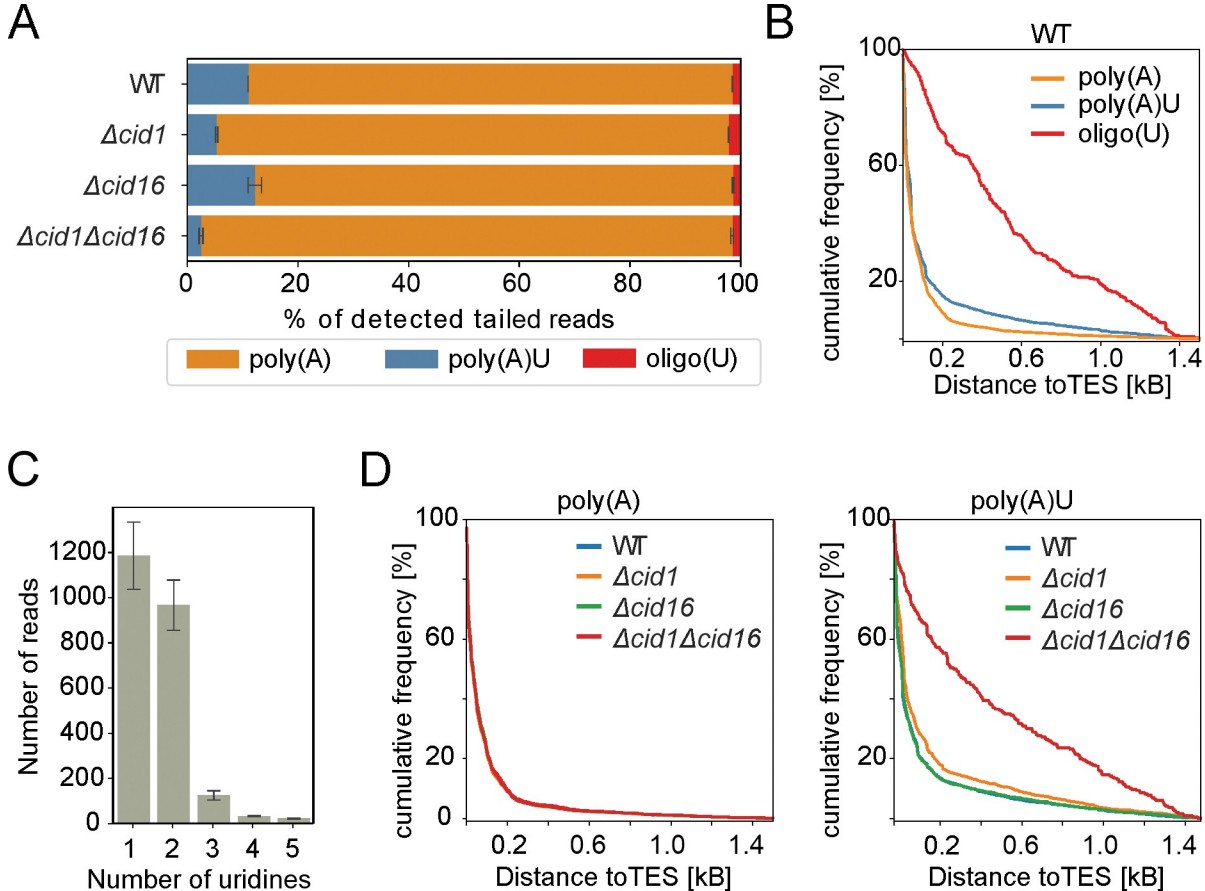

**Fig 2. Both fission yeast uridyltransferases are implicated in mRNA uridylation. (A)** Four categories of reads with A/U extensions that mapped to mRNA were identified in wild-type yeast (WT) and in single or double mutants of two annotated terminal uridyltransferases (TUT-ases)–*cid1* and *cid16*. While *cid1* deletion resulted in a significant drop in uridylation, the double mutant showed a further decrease in the number of detected uridylated reads. Average of two repeats for each sample with standard deviation is shown. **(B)** In wild-type cells, reads with poly(A) and poly(A)U extensions map predominantly to annotated transcription end site (TES) while oligo(U) reads have a looser association with TES. Distance of reads from TES is shown as cumulative frequency. **(C)** Detected poly(AU) extensions carried predominantly one or two uridines. Average of two repeats of wild type sample with standard deviation is shown. **(D)** Reads with poly(A) extensions mapped predominantly to TES in all tested mutants, while reads with poly(AU) extensions in strain with both TUT-ase deleted lost association with TES suggesting the artefactual origin of those reads.

extensions are the ones that are highly abundant in the sample (Fig 3C) and are enriched in highly expressed genes categories (Fig 3D). We thus conclude that uridylation is pervasive in fission yeast while its detection by our protocol is limited only by sequencing depth.

We observed a strong positive correlation between the number of reads with poly(A) or poly(A)U extensions detected in WT and *Δcid16* samples (Fig 3E). There was a somewhat lower correlation between reads with poly(A) and reads with poly(A)U extensions both within and between samples (Fig 3E). This difference might originate from different poly(A) tail lengths of different transcripts or different uridylation frequencies per transcript.

## Deletion of *cid16* but not of *cid1* leads to subtle changes in RNA steady-state levels

We observed that neither the single nor double TUT-ase deletion mutants showed significant impact on fission yeast growth in standard conditions (Fig 4A). We next checked if phenotype of loss of uridylation can be observed when cells are confronted with unfavourable conditions

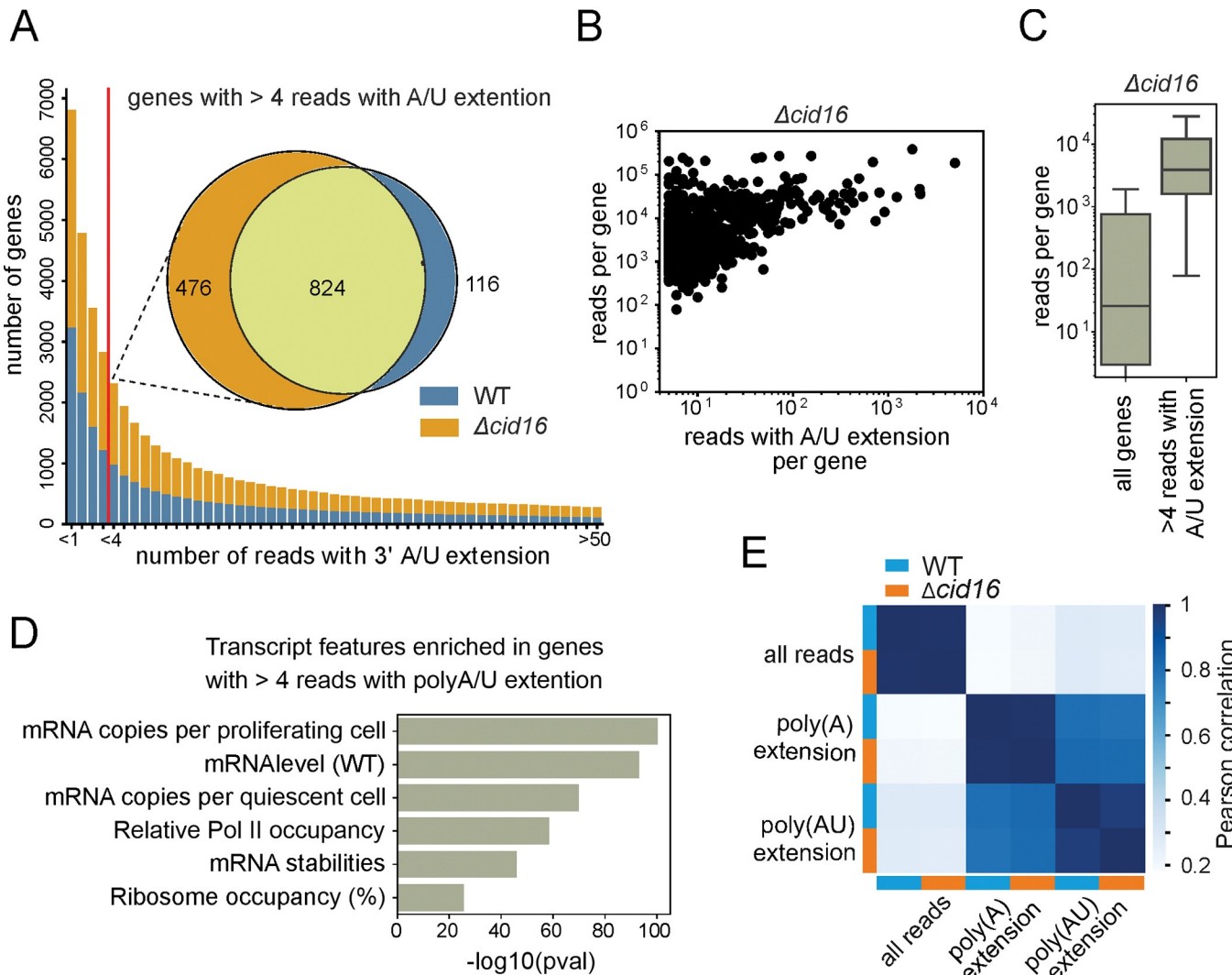

**Fig 3. Uridylation is pervasive in fission yeast. (A)** The number of genes with different amounts of reads with A/U extensions were plotted for two strains with high uridylation frequency (WT and *Δcid16*). Each bar represents the number of genes with at least as many reads detected. Avarage of two repeats with standard deviation is shown for each strain. Venn diagram depicts overlap between genes with more than 4 A/U tailed reads detected in both repeats of each strain. **(B)** The number of reads with un-templated extension positively correlates with gene expression level. Depicted are average read counts for transcripts in *Δcid16* mutant plotted against the sum of reads with A/U extension. Only transcripts with more than 4 reads with A/U extensions in two repeats are shown. **(C)** More reads with un-templated extensions were detected for genes that are highly expressed in investigated samples. Depicted are data for *Δcid16* mutant **(D)** Transcript features of genes for which more than 4 reads with un-templated extensions were detected confirm that we observe predominantly highly expressed genes (data for *Δcid16* mutant). **(E)** Heat map of Pearson correlation calculated for different read categories detected in wild-type sample and *Δcid16* mutant–all reads per transcript, reads with poly(A) extension and reads with uridylated poly(A) extension (poly(AU)). Data for 857 genes that have more than 4 reads with A/U extensions detected in both WT and *Δcid16* samples (as shown at 3A Venn Diagram).

that induce general stress responses [19]. All examined strains grew similarly under oxidative stress, heat shock, respiratory conditions, in presence of respiration inhibitors, translation inhibitors and on different nitrogen sources (S1 Fig). We also checked sensitivity for hydroxy-urea and caffeine [20] described earlier for *cid1* deletion, however, in our hands' none of the tested strains exhibited sensitivity to combined or separate agents (S2 Fig). This could be due to different media and genetic backgrounds used in the previous study. Overall we concluded that uridylation in fission yeast is dispensable for growth control in normal conditions and is not required to stress resistance.

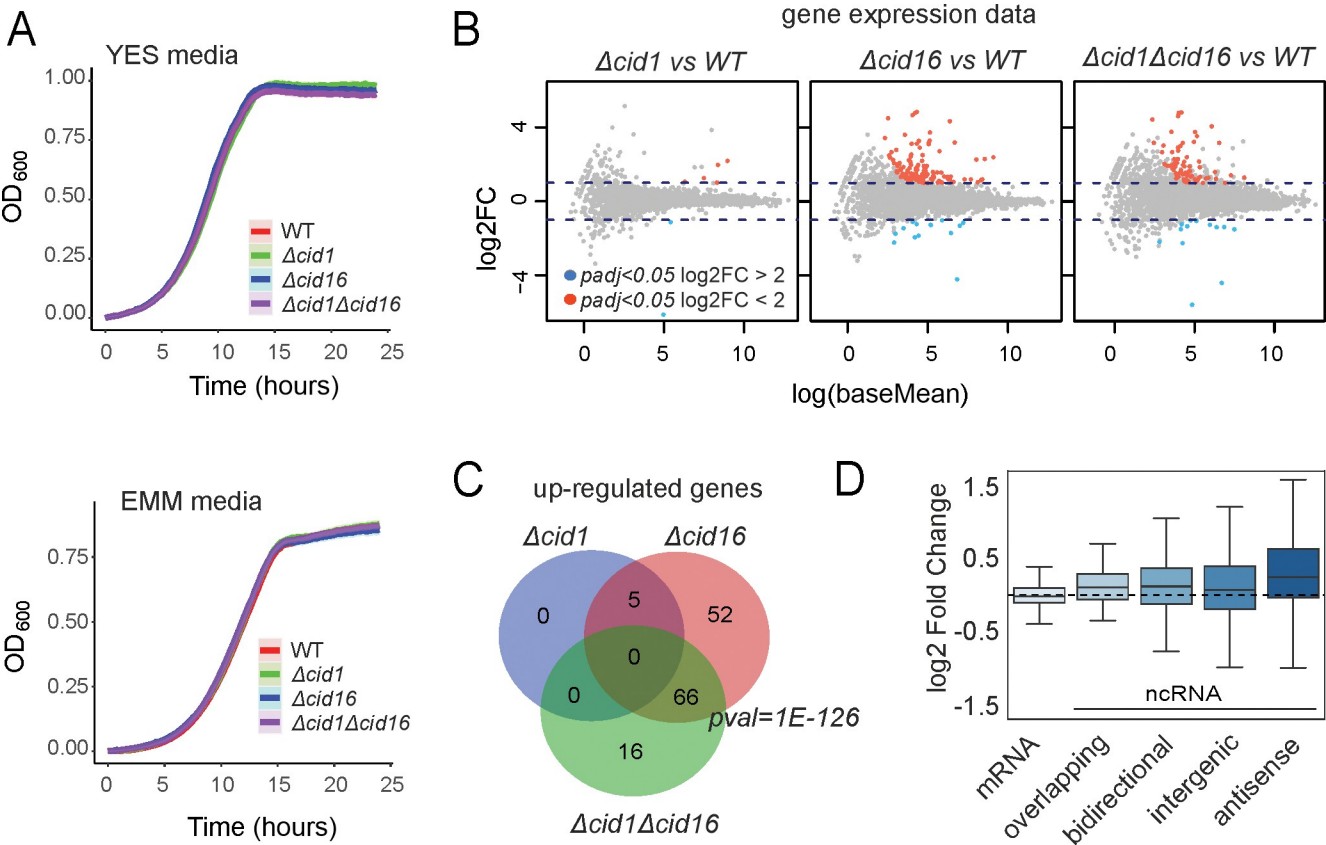

**Fig 4. Characterisation of strains with TUT-ase deletions. (A)** Deletion of each or both fission yeast annotated terminal uridyltransferase genes, *cid1* and *cid16*, have no detectable effect on growth. The growth of strains with deleted genes in complete (YES) and synthetic media (EMM) was recorded using Bioscreen C equipment. Growth curves represent an average of at least four repeats. **(B)** *Cid16* but not *cid1* deletion induces mild changes in gene expression. Genes expression in different TUTase deletion strains grown in YES media was compared to gene expression of the wild-type cells (WT) using DESeq2 workflow [21]. Marked red and blue on the MA plot are genes with an adjusted *p-value* below 0.05 and fold change greater than 2 (red) or smaller than 0.5 (blue) compared to the wild type strain. **(C)** Overlap between up-regulated genes in all three investigated strains indicate that *cid16* deletion is the main cause of observed changes. **(D)** Boxes on the plot represent changes in expression of different gene categories in *Δcid16* strain compared to wild type (WT) sample. The highest overall change was observed for antisense non-coding RNAs.

We next used our RNA sequencing data to check if eliminating uridylation results in significant changes in transcript levels. We used the standard DESeq2 pipeline to identify Differentially Expressed Genes (DEGs) in the analysed mutants compared to the wild-type strain. In agreement with the lack of growth phenotype, we observed only subtle differences between gene expression of wild-type cells and *cid1* deletion mutant, with only 7 significant DEGs between strains (*p-value* < 0.05; fold change > 2) (Fig 4B) (S1 Table). We found more DEGs in *Δcid16* and double deletion strain (137 and 95 DEGs, respectively). In both strains with *cid16* deletion, we observed weak up-regulation of similar groups of genes (Fisher's test *p-value* < 2.2e-16, Fig 4C). Genes up-regulated in deletion of *cid16* or both *cid1* and *cid16* were enriched in long non-coding RNAs with 62 lncRNAs out of 123 up-regulated features in *Δcid16* strain (*p-value* = 1.3E-09), and 44 lncRNAs out of 82 up-regulated features in *Δcid1Δcid16* strain (*p-value* = 1.8e-07, AnGeLi) [22] (S1 Table). We noticed that up-regulated RNAs were predominantly antisense to other features. We used recent annotation to divide non-coding genomic features into categories: antisense, bidirectional, intergenic, and overlapping ncRNAs [23]. A comparison of overall expression changes for each category showed a general trend of antisense ncRNA upregulation in *Δcid16* (Fig 4D).

# Discussion

Uridylation of mRNAs has been reported over a decade ago, since than involvement of uridylation in several specific cellular processes was demonstrated. Notable examples of specific roles for uridylation are regulation of transcriptome changes during germ lines development [10, 11], targeting decay by-products of apoptosis triggered mRNA decay [12], or targeting viral RNAs [8]. In all those cases uridylation triggers decay of its targets. However, mRNAs are pervasively uridylated also in standard growth conditions. It is widely accepted that this widespread uridylation supports mRNA turnover [4]. It was recently shown that elimination of uridylation in plants leads to global mRNA tail length shortening which may impact its downstream processing. Moreover *A. thaliana* uridyltransferase URT1 was reported to directly interact with decapping complex suggesting its integral role in mRNA turnover pathway [13]. Surprisingly elimination of uridylation has little or no effect on organisms' physiology in standard conditions, moreover example of budding yeast proves that organism can cope perfectly with mRNA decay without uridylation. Therefore, the question remains why widespread uridylation is beneficial and evolved in almost all eukaryotic organisms.

Simple eukaryotic model may be very useful to establish exact role of uridylation in mRNA decay. We show here that transcript uridylation in fission yeast shares most of characteristics with other species–uridylation is pervasive, it is detected almost exclusively on poly(A) tails, and the added U extensions are short. We demonstrated that, while Cid1 is the main fission yeast TUT-ase, both Cid1 and Cid16 can contribute to mRNA uridylation.

Our results suggest that Cid16 regulates non-coding antisense transcription. This could be an effect of the reported function of Cid16 in uridylation of Argonaute bound small RNAs [24], which in turn could impact chromatin silencing and antisense transcription. However, due to limited number of repeats and only subtle changes uncovered in our data set such conclusion should be taken with caution. Moreover, *cid16* gene is positioned next to *mlt1* gene which was shown to regulate non-coding and meiotic transcripts [25], and with our experimental set-up we cannot exclude some interference between *cid16* deletion cassette and *mlt1* expression.

While the role of Cid16 in control of specific transcript cannot be established, our results clearly indicate that Cid16 together with Cid1 participate in mRNA uridylation in *S. pombe*. We postulate that fission yeast can be a powerful unicellular model to study the significance of mRNA uridylation, and our data and methods lay a solid foundation for such future research.

Importantly, we demonstrate that transcript uridylation can be detected genome-wide using simple means by analysing RNA sequencing data originating from protocols preserving the 3'-ends of RNAs. To our knowledge the only method that can reliably asses uridylation genome-wide is TAIL-seq [16]. Because our data originates from fragmented reads, we cannot make assumptions regarding actual tail length or measure actual uridylation frequency which is possible using more sophisticated TAIL-seq method. From the other hand, straightforward approach we propose here facilitates fast identification of uridyltransferases involved in pervasive RNA uridylation by comparing changes in detected global uridylation. This approach can be easily adapted to other organisms to identify uridyltransferases responsible for pervasive uridylation.

# Materials and methods

## Strains and media

*S. pombe* 972 *h-* strain was used as a control and to create the deletion mutants Δ*cid1*, Δ*cid16*, and Δ*cid1*Δ*cid16*. *S. pombe* cells were grown in standard conditions and media (32˚C in yeast

extract with supplements (YES), or in Edinburgh minimal medium (EMM). To monitor respiratory growth, cells were cultivated in YE media with 2% glycerol and 0.1% glucose as carbon source. For screening responses to stressors, media were supplemented with antimycin A, chloramphenicol, CsCl, cycloheximide, ethanol, G418, $H_2O_2$, hygromycin, KCl, $MgCl_2$, NaCl, sorbitol, arginine, glutamic acid, methionine or proline (the details are provided in S1 Table.). For the heat-shock, exponentially growing cells were incubated at a temperature of 40 or 50˚C for 10 or 20 minutes before re-starting growth at standard temperature. Experiments were carried out in two biological and at least three technical replicates.

## Growth analysis

The growth curves were obtained by monitoring changes in optical density (OD 600) using micro-bioreactor Bioscreen C. Cells were grown at 32˚C in 100-well plates in 100 μL volume. Exponentially growing pre-cultures were used to start growth in the plate. The initial OD 600 of the culture was set to 0.10, and optical density was measured every 15 minutes for 48 hours.

The growth curves were extracted from the data using Pyphe growth curves Python module [26]. The maximum slope of the growth curve (describing the growth rate of the culture) and the lag phase (the time required for doubling the initial biomass) were determined. For heat map, results of the maximum slope and lag phase of deletion mutants were normalized to wild-type strain for each condition, independently. Statistical analysis was done with one-way ANOVA using Python (SciPy library). Growth curves were visualised using a custom shiny app (https://michamaleckis.shinyapps.io/growth_curves_app/).

## RNA-sequencing

Wild-type cells (*972 h-*) of *S. pombe* and TUT-ase deletion strains in two biological repeats were grown in YES at 32˚C to the early exponential growth phase (OD 0.5). Total RNA was isolated using the hot phenol method. RNA quality was assessed on a Bioanalyzer instrument (Agilent), treated with DNase (Turbo DNA-free, Ambion) and subsequently, 4 μg of RNA was treated with a beta version of Ribo-Zero Magnetic Gold Kit Yeast (Epicentre) to deplete rRNAs. RNA-seq libraries were prepared from rRNA-free RNA using a strand-specific library preparation protocol (a customised version of Illumina TrueSeq Small RNA) [18] and sequenced on an Illumina HiSeq instrument (126-nt paired-end reads).

Reads were checked for quality with FastQC, version 0.11.3. Adapters were removed using Cutadapt version 2.10 from Read 1 and Read 2, respectively, and then, BBMap was implemented to repair disordered reads (BBMap repair function). Reads 1 and 2 were mapped to the *S. pombe* 972 h- reference genome using HiSat2 version 2.1.0.

Count matrices were generated from BAM files using HTSeq version 0.11.1. Differentially expressed genes (DEGs) were determined using DESeq2 version 1.24.0 [21] (S1 Table). Based on the DESeq2 results, lists of down- and upregulated genes were prepared for each strain, and enrichments were investigated using AnGeLi (Analysis of Gene Lists) [22] with a threshold of p-value equal to 0.01 (Fisher's exact test for count data).

## Identification of 3' mRNA extensions in RNA-seq data

The composition, length and location of 3' non-templated extensions of sequencing reads were identified according to soft-clipping information encoded in the ".sam" file CIGAR string. At first, we selected only primary aligned reads in the "sam" file. Then, the file was converted into "bed" format, and the closest tool (bedtools) was used to find overlapping features in the data and the reference (reads were assigned to genes). We next created a table with read sequence, tail sequence, CIGAR string and feature where given read aligns and mapping

positions. This table was used to use CIGAR string information to extract potential 3'-end non-templated extension—part of the read sequence of length S (based on CIGAR) was copied from the 3' end (based on strandness). In the final step, we restricted tails to ones containing only A/U stretches and we categorised them as poly(A)-, poly(A)U-, oligo(U)-, or mixed-tailed, and measured the length of the tail. All pipeline is available in the form of Jupiter note-book in our GitHub repository (https://github.com/igib-rna-tails/Detect_Tails_from_Cigar).

The raw data file containing reads with 3'-A/U extensions was deposited in Sequence Read Archive (SRA, NCBI) under accession number: PRJNA862652. In most of the analyses, we only considered reads with extensions longer than 3 nucleotides as tailed.

## Sequencing data

Raw ".fastq" files from RNA sequencing were deposited in in Sequence Read Archive (SRA, NCBI) under accession number: PRJNA862652. The raw data file containing reads with 3'-A/U extensions was deposited in OSF database - https://osf.io/em7vp/.

## Supporting information

**S1 Fig. Comparison of growth of wild type, *Δcid1* and *Δcid16* strains in different conditions.**
(PDF)

**S2 Fig. Sensitivity of wild type, Δcid1 and Δcid16 strains to hydroxyurea and caffeine.**
(PDF)

**S1 Table. Supporting table.** Table content section S1: number of reads with polyA or poly(A) U extensions detected in analysed samples; section S2: stress conditions used in growth experiment (S1 Fig); section S3: differentially expressed genes list; section S4: functional categories significantly enriched or underrepresented in lists of differentially expressed genes; section S5: summary statistics of RNA sequencing experiment.
(XLSX)

## Acknowledgments

We would like to thank Szymon Świeżewski for fruitful discussions and crucial suggestions regarding data analysis.

## Author Contributions

**Conceptualization:** Michał Małecki.

**Formal analysis:** Lidia Lipińska-Zubrycka, Maciej Grochowski, Michał Małecki.

**Funding acquisition:** Lidia Lipińska-Zubrycka, Jürg Bähler, Michał Małecki.

**Investigation:** Lidia Lipińska-Zubrycka, Maciej Grochowski.

**Methodology:** Lidia Lipińska-Zubrycka.

**Project administration:** Michał Małecki.

**Resources:** Jürg Bähler.

**Software:** Lidia Lipińska-Zubrycka.

**Supervision:** Michał Małecki.

**Validation:** Michał Małecki.

**Visualization:** Michał Małecki.

**Writing – original draft:** Michał Małecki.

**Writing – review & editing:** Lidia Lipińska-Zubrycka, Maciej Grochowski, Jürg Bähler, Michał Małecki.

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
