## [Decision Letter · Decision Letter 0]

14 Mar 2023

PONE-D-23-03513Pervasive mRNA uridylation in fission yeast is catalysed by both Cid1 and Cid16 terminal uridyltransferases.PLOS ONE

Dear Dr. Malecki,

Thank you for submitting your manuscript to PLOS ONE. After careful consideration, we feel that it has merit but does not fully meet PLOS ONE’s publication criteria as it currently stands. Therefore, we invite you to submit a revised version of the manuscript that addresses the points raised during the review process.

We look forward to receiving your revised manuscript.

Kind regards,

Thomas Preiss, PhD

Academic Editor

PLOS ONE

Journal Requirements:

   "This work was funded by EU Operational Programme Innovative Economy via the Foundation for Polish Science grant FIRST TEAM awarded to MM (POIR.04.04.00-00-4316/17). This work was supported by Wellcome Senior Investigator Award to JB (095598/Z/11/Z). This work was supported by National Science Center MIniatura grant (2019/03/X/NZ2/00787) awarded to LL-Z"

Additional Editor Comments:

The two reviewers suggested several specific improvements to your manuscript that I would expect you to implement in revision.

Reviewers' comments:

Reviewer's Responses to Questions

**Comments to the Author**

1. Is the manuscript technically sound, and do the data support the conclusions?

Reviewer #1: Partly

Reviewer #2: Yes

2. Has the statistical analysis been performed appropriately and rigorously? 

Reviewer #1: Yes

Reviewer #2: Yes

3. Have the authors made all data underlying the findings in their manuscript fully available?

Reviewer #1: Yes

Reviewer #2: Yes

4. Is the manuscript presented in an intelligible fashion and written in standard English?

Reviewer #1: Yes

Reviewer #2: Yes

5. Review Comments to the Author

Reviewer #1: Previous studies established that mRNA 3’ uridylation is pervasive in diverse eukaryotes and that this post-transcriptional modification seems to play a role in mRNA turnover, in part through engagement of the 3’-5’ exonuclease DIS3L2. In the fission yeast Schizosaccharomyces pombe mRNA 3’ uridylation was shown to be mostly dependent on the Cid1 uridyltransferase, though a background level of uridylation activity in strains deleted for Cid1 suggested the presence of a second uridyltransferase (Rissland, 2007; 2009). Here, the authors use bulk sequencing of fragmented RNA to demonstrate that this second uridyltransferase is the Cid1 ortholog Cid16, which was previously shown to possess such activity (Preston, 2019). The phenotypic characterisation of mutant strains lacking one or both uridyltransferases suggests that only subtle changes in steady-state mRNA levels result from perturbation of mRNA uridylation in this organism.

The RNA sequencing and bioinformatics aspects of the study appear robust and are clearly described, but the extent to which the findings advance understanding of this area of RNA metabolism is fairly modest; the existence of mRNA uridylation in S. pombe and the RNA uridyltransferase activities of Cid1 and Cid16 were established some years ago. The authors should in any case be encouraged to address the following points:

1 (line 158) Can the authors reasonably conclude that Cid16 activity is only evident in the cid1 deletion background? The data suggest that the deletion phenotypes are not additive, but the contribution of Cid16 to the steady-state level of mRNA uridylation in a wild-type strain cannot easily be determined.

2 The study that first described cid1 showed that the cid1 deletion has specific genetic interactions with mutations in subunits of replicative DNA polymerases (Wang et al. (2000). Mol Cell Biol 20, 3234–3244; strangely not cited here). Why was this particular phenotype not tested or discussed here? Does the handful of genes that are differently expressed on deletion of cid1 or cid16 offer any insight into this genetic interaction?

Reviewer #2: The manuscript “Pervasive mRNA uridylation in fission yeast is catalysed by both Cid1 and Cid16 terminal uridyltransferases” aims to analyze the uridylation of mRNA. The authors showed that uridylation can be detected using a bioinformatics approach. The authors also demonstrated that two terminal uridyltransferases, Cid1 and Cid16, contribute to transcript uridylation. The manuscript is interesting and should be published. However, I hope that the authors revise some following points before the publication.

1) The orange and green boxes should be described the explain in Fig. 1A.

2) Each ratio (%) should be shown in the circle chart of Fig. 1C.

3) The cid1∆ cid16∆ double mutant significantly decreases the uridylation of mRNA but still remains as shown in Fig. 2A. If the authors predict any candidate protein in S. pombe, the additional description is better in the Discussion.

4) The author described “973 transcripts in the wild-type samples and 1,339 in the Cid16 deletion strain samples” (Lines 177-178). However, 940 (824 and 116) in the wild-type and 1,300 (824 and 476) in cid16∆ strains are shown in Fig. 3A. If I am right understanding, the authors should revise this.

5) Typographical errors

Line 180 Change “wt and cid16 sample” to “WT and Cid16 sample”.

Line 187 Not italic for “WT”.

Line 268. Maybe italic for “A. thaliana”.

Line 306. Change “h-” to italic.

6. PLOS authors have the option to publish the peer review history of their article (what does this mean?). If published, this will include your full peer review and any attached files.

Reviewer #1: **Yes: **Chris Norbury

Reviewer #2: No

---

## [Author Response · Author response to Decision Letter 0]

19 Apr 2023

Reviewer #1: Previous studies established that mRNA 3’ uridylation is pervasive in diverse eukaryotes and that this post-transcriptional modification seems to play a role in mRNA turnover, in part through engagement of the 3’-5’ exonuclease DIS3L2. In the fission yeast Schizosaccharomyces pombe mRNA 3’ uridylation was shown to be mostly dependent on the Cid1 uridyltransferase, though a background level of uridylation activity in strains deleted for Cid1 suggested the presence of a second uridyltransferase (Rissland, 2007; 2009). Here, the authors use bulk sequencing of fragmented RNA to demonstrate that this second uridyltransferase is the Cid1 ortholog Cid16, which was previously shown to possess such activity (Preston, 2019). The phenotypic characterisation of mutant strains lacking one or both uridyltransferases suggests that only subtle changes in steady-state mRNA levels result from perturbation of mRNA uridylation in this organism.

The RNA sequencing and bioinformatics aspects of the study appear robust and are clearly described, but the extent to which the findings advance understanding of this area of RNA metabolism is fairly modest; the existence of mRNA uridylation in S. pombe and the RNA uridyltransferase activities of Cid1 and Cid16 were established some years ago. The authors should in any case be encouraged to address the following points:

We would like to thank Reviewer 1 for an overall positive review. While we noted reviewer concerns regarding the modest novelty of our study we believe we are providing useful and interesting information. In particular, we introduce a straightforward strategy to detect uridylation and tools that should allow others to use our approach. Moreover, we facilitate future research on uridylation in fission yeast by demonstrating the pervasiveness of this process and identifying contributing enzymes.

1 (line 158) Can the authors reasonably conclude that Cid16 activity is only evident in the cid1 deletion background? The data suggest that the deletion phenotypes are not additive, but the contribution of Cid16 to the steady-state level of mRNA uridylation in a wild-type strain cannot easily be determined.

After consideration, we decided to remove the mentioned statement from the manuscript. Indeed, based on our data it is not possible to describe the contribution of each of the TUT-ases to mRNA uridylation in the wild-type fission yeast and our statement may sound like we hypothesise about it. However, the difference in uridylation frequency observed in the cid1 deletion strain compared to the double deletion strain allows us to conclude that Cid16 can target mRNA.

We believe that this finding is important for further study of uridylation in fission yeast. While the activity of Cid16 as terminal uridyltransferase was shown previously (Preston, 2019), it was implicated in the turnover of Argonaut-bound small RNAs (Pisacane, 2017) and not in mRNA metabolism.

2 The study that first described cid1 showed that the cid1 deletion has specific genetic interactions with mutations in subunits of replicative DNA polymerases (Wang et al. (2000). Mol Cell Biol 20, 3234–3244; strangely not cited here). Why was this particular phenotype not tested or discussed here? Does the handful of genes that are differently expressed on deletion of cid1 or cid16 offer any insight into this genetic interaction?

We appreciate this comment. We reported no obvious phenotype of TUT-ases deletion strains in several tested conditions. Previously it was reported that the cid1 deletion strain was especially sensitive to a mix of ribonucleotide reductase inhibitor hydroxyurea (HU) and kinase inhibitor caffeine (Wang, 200). While overexpression of cid1 was shown to suppress the HU sensitivity of polymerase mutants.

We included those conditions in our initial screen of deletion strains, however, the readout from liquid media was not conclusive (not shown) and we did not include it in the final manuscript. We repeated those experiments now using plate assays and added a Supplementary Figure to the manuscript.

Interestingly we did not detect the reported sensitivity of cid1 deletion on HU caffeine plates, even so, we see the overall negative effect of the treatment. This information was added to the manuscript as well as a citation of Wang et al. 2000.

Lines 222-226

“We also checked sensitivity for hydroxyurea and caffeine described earlier for cid1 deletion (20), however, in our hands none of the tested exhibited sensitivity to combined or separate agents (S2 Figure). This could be due to different media and genetic backgrounds used in previous study.”

We speculate that the discrepancy in the results may be because the original assay was done using auxotrophic strains while we used deletion in the prototroph background. Our unpublished data show that uridylation frequency is highest for transcripts of amino and glucose metabolism proteins. For this discussion, we can speculate that the effect of cid1 deletion may be somewhat enhanced or modulated by auxotrophies, even in supplemented media (Alam et al. 2016, Nat Microbiol).

Genes under or over-expressed in the studied mutants were checked for GO categories enrichment and double-checked by us. We did not find any significant clues suggesting that cid1 deletion could impact levels of transcripts of genes implicated in replication.

Reviewer #2: The manuscript “Pervasive mRNA uridylation in fission yeast is catalysed by both Cid1 and Cid16 terminal uridyltransferases” aims to analyze the uridylation of mRNA. The authors showed that uridylation can be detected using a bioinformatics approach. The authors also demonstrated that two terminal uridyltransferases, Cid1 and Cid16, contribute to transcript uridylation. The manuscript is interesting and should be published. However, I hope that the authors revise some following points before the publication.

We would like to thank the reviewer for their positive opinion and for the suggestions that will improve our manuscript. 

1) The orange and green boxes should be described the explain in Fig. 1A.

A description was added to the figure

2) Each ratio (%) should be shown in the circle chart of Fig. 1C.

The % ratio information was added to the pie chart.

3) The cid1∆ cid16∆ double mutant significantly decreases the uridylation of mRNA but still remains as shown in Fig. 2A. If the authors predict any candidate protein in S. pombe, the additional description is better in the Discussion.

We suggested in line 167 that residual uridylation detected in double mutant is most probably an artefact of the method.

4) The author described “973 transcripts in the wild-type samples and 1,339 in the Cid16 deletion strain samples” (Lines 177-178). However, 940 (824 and 116) in the wild-type and 1,300 (824 and 476) in cid16∆ strains are shown in Fig. 3A. If I am right understanding, the authors should revise this.

We thank the Reviewer 2 for noticing this, indeed this was a mistake. The values on the figure are right and we corrected the numbers in the text accordingly.

5) Typographical errors

Line 180 Change “wt and cid16 sample” to “WT and Cid16 sample”.

Line 187 Not italic for “WT”.

Line 268. Maybe italic for “A. thaliana”.

Line 306. Change “h-” to italic.

We thank the Reviewer 2 for spotting those errors. All were corrected in the text.

---

## [Decision Letter · Decision Letter 1]

26 Apr 2023

Pervasive mRNA uridylation in fission yeast is catalysed by both Cid1 and Cid16 terminal uridyltransferases.

PONE-D-23-03513R1

Dear Dr. Malecki,

We’re pleased to inform you that your manuscript has been judged scientifically suitable for publication and will be formally accepted for publication once it meets all outstanding technical requirements.

Kind regards,

Thomas Preiss, PhD

Academic Editor

PLOS ONE

Additional Editor Comments (optional):

Reviewers' comments:

Reviewer's Responses to Questions

**Comments to the Author**

1. If the authors have adequately addressed your comments raised in a previous round of review and you feel that this manuscript is now acceptable for publication, you may indicate that here to bypass the “Comments to the Author” section, enter your conflict of interest statement in the “Confidential to Editor” section, and submit your "Accept" recommendation.

Reviewer #1: All comments have been addressed

Reviewer #2: All comments have been addressed

2. Is the manuscript technically sound, and do the data support the conclusions?

Reviewer #1: Yes

Reviewer #2: Yes

3. Has the statistical analysis been performed appropriately and rigorously? 

Reviewer #1: Yes

Reviewer #2: Yes

4. Have the authors made all data underlying the findings in their manuscript fully available?

Reviewer #1: Yes

Reviewer #2: Yes

5. Is the manuscript presented in an intelligible fashion and written in standard English?

Reviewer #1: Yes

Reviewer #2: Yes

6. Review Comments to the Author

Reviewer #1: (No Response)

Reviewer #2: (No Response)

7. PLOS authors have the option to publish the peer review history of their article (what does this mean?). If published, this will include your full peer review and any attached files.

Reviewer #1: **Yes: **Chris Norbury

Reviewer #2: No

---

## [Editor Report · Acceptance letter]

12 May 2023

PONE-D-23-03513R1 

Pervasive mRNA uridylation in fission yeast is catalysed by both Cid1 and Cid16 terminal uridyltransferases. 

Dear Dr. Malecki:

I'm pleased to inform you that your manuscript has been deemed suitable for publication in PLOS ONE. Congratulations! Your manuscript is now with our production department. 

Kind regards, 

on behalf of

Prof Thomas Preiss 

Academic Editor

PLOS ONE